# Anti-Obesity Effects of the Larval Powder of Steamed and Lyophilized Mature Silkworms in a Newly Designed Adult Mouse Model

**DOI:** 10.3390/foods12193613

**Published:** 2023-09-28

**Authors:** Min Woo Kim, Yu-Jin Ham, Hyun-Bok Kim, Ji young Lee, Jung-Dae Lim, Hyun-Tai Lee

**Affiliations:** 1Biopharmaceutical Engineering Major, Division of Applied Bioengineering, College of Engineering, Dong-eui University, Busan 47340, Republic of Korea; diceandlife@gmail.com (M.W.K.); keivn2076@naver.com (Y.-J.H.); 2National Institute of Agricultural Sciences, Rural Development Administration, Wanju 55365, Republic of Korea; hyunbok@korea.kr; 3Department of Ophthalmology and Visual Science, Daejeon St. Mary’s Hospital, College of Medicine, The Catholic University of Korea, Seoul 06591, Republic of Korea; ram1020@naver.com; 4Department of Herbal Medicine Resource, Kangwon National University, Samcheok 25949, Republic of Korea; ijdae@kangwon.ac.kr; 5Core-Facility Center for Tissue Regeneration, Dong-Eui University, Busan 47340, Republic of Korea

**Keywords:** adipose tissue, appetite, body weight, *Bombyx mori*, high-fat diet, food intake, mature silkworm, normal diet, nutraceutical, obesity

## Abstract

Recently, “mature” silkworms (MS) of *Bombix mori* have been considered a potential nutraceutical, with a number of health benefits reported for steamed and lyophilized MS powder (SMSP). However, no obesity-related effects have been reported for SMSP. In the present study, anti-obesity effects of SMSP were investigated in adult mice in vivo, aged 12 weeks at the onset of SMSP treatment, fed a normal diet (ND) and a high-fat diet (HFD), respectively, for 12 weeks. SMSP significantly suppressed body weight gain, intra-abdominal adipose tissue, and food efficiency in both ND-fed and HFD-fed adult mice. In addition, SMSP significantly decreased food intake and liver weight in HFD-fed mice, indicating that SMSP suppressed appetite and simultaneously reduced the conversion of feed into body weight in HFD-fed mice. SMSP also significantly lowered the serum levels of glucose, triglyceride, total cholesterol, low-density lipoprotein cholesterol, asparagine transaminase, alanine transaminase, and alkaline phosphatase. However, SMSP had no significant effect on the weights of the kidney, spleen, or thymus or the serum levels of blood urea nitrogen and creatinine. Taken together, the above results suggest that SMSP has potent anti-obesity effects and is safe for long-term use as a potential therapeutic and/or nutraceutical in both obese patients and non-obese individuals.

## 1. Introduction

Mulberry silkworms, *Bombyx mori* L., and their by-products have provided human beings with a variety of dietary supplements, as well as their cocoons being used for making fabrics for thousands of years [1]. As a sericulture product, “mature” silkworms (MS), which are silkworms on the 7th or 8th day of the 5th instar larvae, have degenerated internal organs and enlarged silk glands enriched with diverse functional nutrients [2]. However, MS have long been unavailable as a nutraceutical because cooked or freeze-dried MS became too hard (mainly due to their denatured silk glands) for humans to chew or digest [3]. A special steaming method for processing edible MS has made the MS a potential health supplement for humans [4]. Indeed, several health-promoting effects of steamed and freeze-dried MS powder (SMSP) have recently been reported in animal models, such as promoting lifespan and health span [2,5,6,7], enhancing memory [7,8], and preventing the onset of Parkinson’s disease [5,6,9], ethanol-induced gastric ulcers [10] and liver disease [11,12], and hepatocellular carcinogenesis [13,14].

In addition, our previous studies have shown that long-term administration of SMSP resulted in skin whitening [15] and hair growth promotion [16] in mice in vivo. While conducting these experiments, we unexpectedly observed that the average body weight (BW) of the SMSP-treated mice was significantly lower than that of the control mice. As is well known, obesity has been regarded as one of the most critical health-related issues worldwide in this century. Although numerous R&D achievements have been reported on obesity and its treatment to date, problems associated with obesity have not yet been fully addressed. Several studies have reported the inhibitory effects of silkworms and silkworm-related sericultural products, such as silkworms on the 3rd day of the 5th instar larvae [17], silk peptide (SP) [18], silk sericin [19,20], silkworm pupa peptide [18,21], and mulberry leaves [22,23] and fruits [24], on obesity and/or BW gain (BWG). However, obesity-related activities have not yet been reported for MS or SMSP.

As an animal model of experimentally induced obesity, mice fed a high-fat diet (HFD) have been widely used in obesity-related studies such as hyperlipidemia and hyperglycemia. However, from both the “quality of life” and “pharmaceutical and food industry” perspectives, nutraceuticals targeting large numbers of non-patients, who are routinely interested in weight control, are just as important as pharmaceuticals for patients with hyperlipidemia and hyperglycemia. In typical in vivo animal experiments, animals are housed under conditions where they have free access to a normal diet (ND), which is similar to the environmental conditions for humans in the modern era, where food intake (FI) is not restricted if desired. Therefore, we have hypothesized that feeding ND instead of HFD to laboratory animals might more closely represent the dietary patterns of normal, appetite-driven individuals who are routinely concerned with weight control, rather than obese patients.

In the previous obesity-related in vivo animal studies, the age of the mice at the start of long-term drug treatment varied from study to study. In many studies, in fact, long-term administration of test agents was initiated in mice younger than 10 weeks of age [19,20,21,22,23,24,25,26,27,28]. Adulthood in animals is biologically defined as the age at which sexual maturity is attained. In mice, sexual maturity is known to be attained at 8–12 weeks of age, with an average of 10 weeks [29]. If so, it could be interpreted that in many previous studies with mice, the in vivo experiments started before the mice had fully reached adulthood. As is well known, however, obesity is one of the leading causes of many adult diseases, so anti-obesity medications and nutraceuticals are primarily targeted at people in adulthood rather than adolescence.

In the present study, therefore, the anti-obesity effects of SMSP were examined under conditions of long-term feeding of ND and HFD, respectively, to adult mice in vivo. Our aims were to investigate the effects of SMSP primarily on BWG and intra-abdominal adipose tissue (IAT) in mice and to compare to those in SP, which has been positioned as one of the major sericultural products and has a composition quantitatively and qualitatively similar to that of SMSP. First, to minimize inter-individual variation in BW changes, 88 5-week-old mice were fed either ND or HFD until they reached 12-week-old adulthood, and then 32 adult mice with similar BW were finally selected at the onset of test agent treatment. This selection process was conducted for both the ND-feeding and HFD-feeding experiments. BW and FI were measured weekly in mice fed ND and HFD, respectively, during the 12-week period of the treatment of test agents. At the end of each 12-week experiment, serum biochemical examinations and measurements of weights of IAT and intra-abdominal organs at necropsy were performed to elucidate the anti-obesity effects and safety of SMSP for its long-term use in adult mice.

## 2. Materials and Methods

### 2.1. Drugs and Reagents

Catechin, Folin–Ciocalteu reagent, orlistat, and quercetin were purchased from Sigma-Aldrich Co. (St. Louis, MO, USA). Methylcellulose (MC) was purchased from Shanghai Aladdin Bio-Chem Technology Co., Ltd. (Shanghai, China). SP, which was attained through the enzymatic degradation of silk proteins [18], was obtained from Worldway Co., Ltd. (Jeoneui, Republic of Korea). The other reagents used in this study were of analytical grade or better and were used without further purification.

### 2.2. Preparation of SMSP

The SMSP used in this study was obtained according to a previously described method [16]. In brief, silkworm larvae of the *Bombyx mori* red cocoon strain (named “Joohwangjam” in Republic of Korea) were raised with ad libitum access to fresh mulberry leaves and were harvested at the National Institute of Agricultural Sciences in Republic of Korea (Wanju, Republic of Korea). Live MS were smothered promptly, steamed at 100 °C for 130 min using an electric pressure-free cooking machine (KumSeong Ltd., Bucheon, Republic of Korea), and lyophilized at −50 °C for 24 h using a freeze-drier (FDT-8612; Operon Co., Ltd., Gimpo, Republic of Korea). The lyophilized MS were then pulverized using a DCH-500D air jet mill (Duksan Co., Ltd., Siheung, Republic of Korea) into SMSP. The diameter of each SMSP particle was around 1.1 μm. A reference specimen (No. BP204) was deposited at the authors’ laboratory in Dong-eui University (Busan, Republic of Korea) for future use.

### 2.3. Quantitative Analysis of Total Polyphenols and Flavonoids in SMSP

To assure the consistent quality of SMSP, the total polyphenols and flavonoids in SMSP were determined using the Folin–Ciocalteu method and the aluminum chloride calorimetric assay, respectively, with minor modifications [15,16]. Catechin was used as a standard compound for the content analysis of the total polyphenols in SMSP. A total of 0.5 g of SMSP was soaked in 30 mL of 80% ethanol with shaking for 20 min, and the suspension was ultrasonicated at 70 °C for 1 h and centrifuged (Centrifuge 5804R, Eppendorf AG, Hamburg, Germany) at 4000 rpm for 15 min. Then, 1 mL of the filtered supernatant (SMSP-E) or a standard solution of catechin was mixed with 1 mL of 1 N Folin–Ciocalteu reagent and 8 mL of distilled water, followed by the addition of 1 mL of 15% Na_2_CO_3_ solution. The mixture was then allowed to stand for 2 h at room temperature, and the absorbance was measured at 760 nm using a UV-1800 Spectrophotometer (Shimadzu, Kyoto, Japan). The amount of total polyphenols was calculated and expressed as the catechin equivalent (i.e., mg of catechin/100 g of SMSP). For the content analysis of the total flavonoids in SMSP, quercetin was used as a standard material. A total of 0.5 mL of SMSP-E (or a standard solution of quercetin) was added to 2 mL of distilled water, followed by the addition of 0.15 mL of 5% NaNO_2_, 0.15 mL of 10% AlCl_3_·6H_2_O, and 1 mL of 1 M NaOH solutions. The mixture was then stirred for 5 min and allowed to stand for 15 min at room temperature, and the absorbance was measured against prepared reagent blank at 510 nm. The amount of total flavonoids was expressed as the quercetin equivalent (i.e., mg of quercetin/100 g of SMSP).

### 2.4. Animals

Five-week-old male ICR mice were obtained from Samtako Bio Korea Co., Ltd. (Osan, Republic of Korea) and were housed individually in separate clear plastic cages at Core-Facility Center for Tissue Regeneration, Dong-eui University (Busan, Republic of Korea). The mice were maintained under conditions of controlled temperature (22 ± 2 °C), humidity (55 ± 5%), and illumination (light on 7 a.m. to 7 p.m.). All animal experiments conducted in the present study were approved by the Institutional Animal Care and Use Committee of Dong-eui University (approval number: R2021-003). Animals were acclimatized to the housing conditions with ad libitum access to a commercial ND (SAM #31, Samtako Bio Korea Co., Ltd., Osan, Republic of Korea) and tap water for two weeks. All animal experiments were conducted between 9 a.m. and 5 p.m.

### 2.5. Experimental Design for ND-Fed Mice

Of the 88 mice adapted to the above housing conditions (see Section 2.4) for 2 weeks (i.e., at 7 weeks of age), 60 mice with BW close to the average BW of the 88 were selected and fed the same ND for an additional 5 weeks. Among the 60 mice fed ND at 12 weeks of age, 32 mice with BW close to the average BW of the 60 were selected and randomly divided into four groups: (i) control; (ii) orlistat 60 mg/kg/day; (iii) SMSP 0.1 g/kg/day; and (iv) SMSP 1 g/kg/day (*n =* 8 in each group). Each test agent was orally administered once daily for 12 weeks. An aqueous suspension of 0.5% MC was used as a vehicle for the “insoluble” SMSP. Orlistat, a representative pancreatic lipase inhibitor used clinically that reduces the absorption of triglycerides (TGs) and the accumulation of fatty acids in the gut, was selected as a positive control drug [22,26,30,31]. A 0.5% MC suspension containing each test agent was administered orally at each dose to ND-fed mice in each group using an orogastric tube at a volume of 10 mL/kg BW. A schematic of the experimental design in ND-fed mice is shown in Figure 1A.

### 2.6. Experimental Design for HFD-Fed Mice

After a 2-week acclimatization period as described in Section 2.4, of the 88 7-week-old mice fed ND, 60 mice with BW close to the average BW of the 88 were selected and fed HFD, a commercial rodent diet containing 60 kcal% fat (D12492, Research Diets Inc., New Brunswick, NJ, USA). After being fed HFD for 5 weeks (i.e., at 12 weeks of age), 32 mice with BW close to the average BW of the 60 were selected from the 60 HFD-fed mice and randomly divided into four groups: (i) control; (ii) orlistat 60 mg/kg/day; (iii) SP 1 g/kg/day; and (iv) SMSP 1 g/kg/day (*n =* 8 in each group). Each test agent was orally administered once daily for 12 weeks. The preparation and administration methods for each test agent were the same as in the ND-feeding experiment (see Section 2.5). A schematic of the experimental design in HFD-fed mice is shown in Figure 1B.

### 2.7. Measurement of BW and FI

BW was measured at the initiation of treatment and once a week thereafter. BWG for each mouse was calculated by subtracting the initial BW from the final BW measured at the end of each 12-week experiment. Each mouse in an individual cage was provided with 30 g of fresh ND or HFD twice a week, and FI was estimated from the weight of the remaining chow. Total FI (TFI) for each mouse was calculated by summing all FI consumed by each mouse over the course of the 12-week experiment. Food efficiency ratio (FER) was calculated as a percentage value of BWG divided by TFI. In addition, mice were carefully inspected daily for any obvious signs of toxicity.

### 2.8. Serum Biochemical Examinations

After measuring final BW at the end of each 12-week experiment (see Section 2.5 and 2.6), each animal was fasted for 20 h with ad libitum access to water. Blood (~0.8 mL) was collected via cardiac puncture from the heart of each mouse under ether anesthesia. At 20 min thereafter, it was centrifuged at 3000 rpm for 10 min. The resultant supernatant (i.e., serum) was then collected for the serum biochemical assessments, which were conducted in the SCL Healthcare Central Laboratory (Yongin, Republic of Korea). Serum levels of glucose, TG, total cholesterol (TC), high-density lipoprotein cholesterol (HDLC), low-density lipoprotein cholesterol (LDLC), blood urea nitrogen (BUN), creatinine, asparagine transaminase (AST), alanine transaminase (ALT), and alkaline phosphatase (ALP) were measured using a Roche cobas c502 analyzer (Roche Diagnostics International Ltd., Rotkreuz, Switzerland) with commercial kits according to the manufacturer’s instructions.

### 2.9. Measurement of Weights of Intra-Abdominal Organs and IAT at Necropsy

After blood collection (see Section 2.8), each animal was sacrificed by cervical dislocation under ether anesthesia and necropsied with special attention paid to all vital organs and tissues after abdominal opening. The entire liver, kidney, spleen, and thymus were excised, and each of the intra-abdominal organs was weighed. IAT was isolated from the epididymal, retroperitoneal, and mesenteric regions [32], and the weight of the isolated IAT was measured.

### 2.10. Statistical Analysis

All data from each experimental group are expressed as the mean ± standard error of the mean (SEM). Data were evaluated using one-way analysis of variance (ANOVA), followed by a Dunnett’s post hoc test, if appropriate. Statistical significance was considered when the *p*-value was less than 0.05.

## 3. Results

### 3.1. Contents of Total Polyphenols and Flavonoids in SMSP

In order to assure the consistency of the quality of SMSP used in the present study, the contents of total polyphenols and flavonoids were measured in the SMSP. The amounts of total polyphenols and flavonoids in 100 g of SMSP were calculated to be 791.6 ± 47.7 mg of catechin (*n =* 4) and 494.3 ± 17.4 mg of quercetin (*n =* 4), respectively, suggesting fairly good consistency in the quality of SMSP in terms of the contents of total polyphenols and flavonoids.

### 3.2. Comparison of the Effects of Feeding ND or HFD to Mice for 5 Weeks

At the onset of the in vivo animal experiments, the average BW of the 60 7-week-old mice selected from the 88 animals in the ND-feeding experiment (Section 2.5) and the HFD-feeding experiment (Section 2.6) was 30.89 ± 0.46 g (*n =* 60) and 30.78 ± 0.48 g (*n =* 60), respectively (Appendix A). After being fed the different diets for 5 weeks, the average BW of the 60 HFD-fed mice (51.56 ± 1.10 g) was significantly (*p* = 4.06 × 10^−11^) higher than that of the 60 ND-fed mice (42.79 ± 0.51 g) (Appendix A), which is consistent with numerous previous studies regarding the effects of HFD on BWG in laboratory animals [17,18,19,20,21,22,23,24,25,26,27,28,30,31].

### 3.3. Effects of SMSP on BW Change and FI in ND-Fed Mice

Information on BW change and food consumption in the four groups of ND-fed mice (i.e., control, orlistat 60 mg/kg, SMSP 0.1 g/kg, and SMSP 1 g/kg) over 12 weeks is shown in Figure 2 and Table 1. Throughout the entire 12-week experiment, no abnormal clinical signs were noted in any of the animals. From week 0 to week 1, there was no significant difference in BW among the four groups. From week 2 through week 12, BWs in the orlistat 60 mg/kg and SMSP 1 g/kg groups were significantly lower than those in the control group, respectively (Figure 2A). There was no significant difference in BW between the orlistat 60 mg/kg and SMSP 1 g/kg groups. BWs in the SMSP 0.1 g/kg group from week 1 to week 12 appeared to be numerically lower than those in the control group, but the difference was not significant in any week (Figure 2A). BWG in each of the orlistat 60 mg/kg and SMSP 1 g/kg groups was significantly lower than in the control group. BWG in the SMSP 0.1 g/kg group was numerically lower than in the control group, but it was not statistically significant (Table 1). Thus, based on the results of the SMSP 0.1 g/kg and 1 g/kg groups, it is evident that SMSP significantly suppressed BWG in a dose-dependent manner. Weekly FIs for 12 weeks and TFIs in the four groups of ND-fed mice are shown in Figure 2B. There was no significant difference in FI or TFI among the four groups, although a slight dose-dependent decrease in both FI and TFI was observed with the administration of SMSP. However, FER, which represents the efficiency of converting feed mass to body mass, was significantly lower in the orlistat 60 mg/kg and SMSP 1 g/kg group, respectively, than in the control group (Table 1).

### 3.4. Effects of SMSP on Weights of IAT and Intra-Abdominal Organs in ND-Fed Mice

The weights of IAT and intra-abdominal organs in the four groups of ND-fed mice are listed in Table 2. Similar to the BWG results, IAT in each of the orlistat 60 mg/kg and SMSP 1 g/kg groups was significantly lighter than in the control group, whereas the SMSP 0.1 g/kg group showed no significant difference in IAT compared to the control group. Photographs of the abdomen and IAT for all 32 ND-fed mice at the end of the experiment are shown in Appendix A. Even with the naked eye, it was discernible that the size of IATs in each of the orlistat 60 mg/kg and SMSP 1 g/kg groups was smaller than in the control group (Appendix A). There were no significant differences in the weights of intra-abdominal organs (i.e., liver, kidney, spleen, and thymus) among the four groups (Table 2).

### 3.5. Effects of SMSP on Serum Biochemical Parameters in ND-Fed Mice

The serum levels of metabolites related with glucose and lipid metabolism in the four groups of ND-fed mice are listed in Table 3. Serum glucose, TG, TC, and LDLC levels in the orlistat 60 mg/kg and SMSP 1 g/kg groups were significantly lower than those in the control group, respectively. Serum HDLC in each of the three agent-treated groups was numerically higher than in the control group, but not statistically significant. Serum levels of glucose, TG, TC, and LDLC in the SMSP 0.1 g/kg group were all slightly lower than in the control group, with no statistically significant differences. Hence, taking the results of the SMSP 0.1 g/kg and 1 g/kg groups together, SMSP significantly reduced serum levels of glucose, TG, TC, and LDLC in a dose-dependent manner (Table 3).

Serum biochemical markers reflecting renal and hepatic toxicity in the four groups of ND-fed mice are listed in Table 4. There were no significant differences in the serum biochemical parameters reflecting renal (i.e., BUN and creatinine) and hepatic (i.e., AST, ALT, and ALP) toxicity among the four groups (Table 4). These results of serum biochemical parameters, together with the results of organ weights (Table 2), suggest the safety of SMSP at high doses (e.g., 1 g/kg) for long-term use in ND-fed mice.

### 3.6. Effects of SMSP on BW Change in HFD-Fed Mice

BW changes in the four groups of HFD-fed mice (i.e., control, orlistat 60 mg/kg, SP 1 g/kg, and SMSP 1 g/kg) over 12 weeks are shown in Figure 3A. No abnormal clinical sign or diarrhea was noted in any of the animals throughout the whole experimental period of 12 weeks, although some mice tended to have loose feces. BWs in the orlistat 60 mg/kg and SMSP 1 g/kg groups were significantly lower than those in the control group from week 2 through week 12, respectively. There was no significant difference in BW between the orlistat 60 mg/kg and SMSP 1 g/kg groups. BWs in the SP 1 g/kg group from week 4 to week 12 were significantly lower than those in the control group. Interestingly, during the final four weeks (i.e., weeks 9 through 12), BWs in the SMSP 1 g/kg group were significantly lower than those in the SP 1 g/kg group (see blue asterisks in Figure 3A). BWG in each of the three agent-treated groups was significantly lower than in the control group (Table 5). In addition, BWG in each of the orlistat 60 mg/kg and SMSP 1 g/kg groups was significantly lower than in the SP 1 g/kg group (see blue asterisks in the BWG row in Table 5), suggesting that SMSP, as a nutraceutical derived from silkworms, is more potent in suppressing BWG than SP.

### 3.7. Effects of SMSP on FI in HFD-Fed Mice

Weekly FI for 12 weeks and TFI in the four groups of HFD-fed mice are shown in Figure 3B. FI was significantly lower in the orlistat 60 mg/kg group at week 1, in the SP 1 g/kg group at weeks 3–5, 8, and 9, and in the SMSP 1 g/kg group at weeks 2–5, 7, 9, and 11, respectively, compared to those in the control group. In addition, FI w significantly lower at weeks 3–5 and 9–12 in the SP 1 g/kg group and at weeks 3–7, 9, 11, and 12 in the SMSP 1 g/kg group, respectively, than those in the orlistat 60 mg/kg group (see gray asterisks in Figure 3B). In line with the results of FI, TFI in each of the SP 1 g/kg and SMSP 1 g/kg groups was also significantly lower than in the control and orlistat 60 mg/kg groups, respectively (see gray asterisks in Figure 3B and the TFI row in Table 5), implying that both SP and SMSP, but not orlistat, might have suppressed the appetite of the HFD-fed mice in this study. FER in each of the three agent-treated groups was significantly lower than in the control group (Table 5). In addition, FER in each of the orlistat 60 mg/kg and SMSP 1 g/kg groups was significantly lower than in the SP 1 g/kg group (see blue asterisks in the FER row in Table 5).

### 3.8. Effects of SMSP on Weights of IAT and Intra-Abdominal Organs in HFD-Fed Mice

The weights of the IAT and intra-abdominal organs in the four groups of HFD-fed mice are listed in Table 6. The IAT and liver weights in the three agent-treated groups were each significantly lower than in the control group. Photographs of the abdomen and IAT for all 32 HFD-fed mice at the end of the experiment are shown in Appendix A. Even with the naked eye, it was quite clear that the size of the IAT in each of the orlistat 60 mg/kg and SMSP 1 g/kg groups was distinctly smaller than in the control group (Appendix A). It is interesting to note that BWG and IAT in the SMSP 1 g/kg group were significantly lower than in the SP 1 g/kg group (see blue asterisks in the BWG and IAT rows in Table 5 and Table 6, respectively), suggesting that SMSP is more potent than SP in suppressing BWG and IAT in HFD-fed adult mice. There were no significant differences in the weights of kidney, spleen, or thymus among the four groups (Table 4), indicating the safety of long-term administration of SMSP at high doses (e.g., 1 g/kg for HFD-fed mice).

### 3.9. Effects of SMSP on Serum Biochemical Parameters in ND-Fed Mice

The effects of SMSP on the serum levels of metabolites related with glucose and lipid metabolism were evaluated in HFD-fed mice (Table 7). In a similar pattern to the ND-feeding experiment (see Table 3), serum levels of glucose, TG, TC, and LDLC in the orlistat 60 mg/kg and SMSP 1 g/kg groups were significantly lower than in the control group, respectively. In the SP 1 g/kg group, glucose, TG, and TC were significantly lower than in the control group, respectively. Serum HDLC in each of the three agent-treated groups was numerically higher than in the control group, but not statistically significant. There was no significant difference in serum glucose, TG, TC, or LDLC among the three agent-treated groups.

Serum biochemical parameters reflecting renal and hepatic toxicity in the four groups of HFD-fed mice are listed in Table 8. There was no significant difference in BUN or creatinine, reflecting nephrotoxicity, among the four groups. Each of the liver enzymes (i.e., AST, ALT, and ALP) in the SMSP 1 g/kg group was significantly lower than in the control group, suggesting that SMSP has hepatoprotective activity.

## 4. Discussion

The present study investigated the anti-obesity effects of SMSP, which has recently gained attention as a potential health supplement containing a variety of functional nutrients for humans, by evaluating BW, FI, weights of IAT and intra-abdominal organs, and serum biochemical parameters in both ND-fed and HFD-fed adult mice in vivo. Our first idea in designing a new animal model for this study was that ad libitum access to ND instead of HFD in mice would more closely represent the dietary patterns of normal, appetitive individuals who are routinely interested in weight control, rather than obese patients. In fact, nutraceuticals targeting non-patients, the majority of the general population, are just as important as pharmaceuticals for obese patients. Second, obesity causes much greater morbidity in adulthood than in adolescence in humans, so it is reasonable to conduct experiments on mice at 12 weeks of age or older, when they have clearly entered adulthood. Third, while mice were housed to full adulthood (i.e., at 12 weeks of age), they underwent a selection process to minimize inter-individual variation in BW changes (see Figure 1).

As previous studies have reported that SMSP is rich in fat [16,33], it is essential to investigate the chemical composition of the fat fraction in more detail. Investigating the impact of the fat fraction on the anti-obesity effects in future studies could provide valuable insights into the mechanism of action. In addition, cellulose, of which there is a high content in plant materials, is known to suppress BWG in HFD-fed mice [25]. However, since SMSP, which is of animal origin, has a crude fiber content lower than 2% [16] and even insects primarily contain chitin as a form of fiber, the anti-obesity effect of the cellulose contained in SMSP need not be considered in this study.

In terms of BW, male mice are known to weigh 20–30 g by the time they reach adulthood [29]. In this study, ND-fed mice had an initial weight of over 40 g (Table 1), and HFD-fed mice over 50 g (Table 5), indicating that all mice reached full adulthood at the onset of test agent treatment. In addition, mice fed HFD for 5 weeks up to that time point had a significant increase in BW compared to mice fed ND (Appendix A). Long-term feeding of HFD for a total of 17 weeks (i.e., 5 weeks of pretreatment followed by 12 weeks of main experiment, see Figure 1) resulted in a significant increase in IAT and liver weights as well as BWG compared to those in the ND-fed mice (Appendix A). Even with the naked eye, it was clear that the IAT of HFD-fed mice was distinctly bigger than that of ND-fed mice (Appendix A). Serum levels of glucose, TG, TC, LDLC, AST, ALT, and ALP were also significantly increased in HFD-fed mice compared to those in ND-fed mice (Appendix A), implying alterations in the metabolism of glucose and lipids along with hepatic damage due to the long-term feeding of HFD. From these results, it could be speculated that the pathophysiologic conditions in HFD-fed and ND-fed mice are substantially different, parallel to the differences between obese patients and non-obese individuals.

Based on final BW, SMSP resulted in approximately 14% and 21% weight loss compared to each control in ND-fed and HFD-fed mice, respectively, which is slightly higher than the efficacy of orlistat in ND-fed mice (~9% weight loss) and comparable in HFD-fed mice (~20% weight loss), but not statistically significant (see Table 1 and Table 5). In both ND-fed and HFD-fed mice, the efficacy of orlistat appeared to be relatively better than that of SMSP in the early weeks of the 12-week experiment, but over time, the efficacy of SMSP appeared to overtake that of orlistat (Figure 2A and Figure 3A). Notably, in ND-fed mice, the BW values of the two groups crossed over and reversed at week 6 and did not reverse again until the end of the experiment (Figure 2A).

Weight gain in the IAT and liver induced by the accumulation of ectopic fat, which is known to cause severe inflammation and organ dysfunction due to an abnormal increase in adipokine secretion, has been considered as one of the important indicators of obesity since excess lipids from dietary intake are stored in intra-abdominal organs, especially in the IAT and liver in obese mice [27,34]. In Figure 4A, SMSP showed a dose dependency for its suppressive effect on BWG and weights of IAT and liver in ND-fed mice, although its effect on liver weight was not significantly different compared to the control group. In HFD-fed mice, SMSP significantly suppressed BWG and weight gain of IAT and liver compared to their respective controls (Figure 4B), suggesting that SMSP suppressed BWG in mice, at least in part, by facilitating the removal of ectopic fat from IAT and the liver. Furthermore, the suppressive effect of SMSP on the increase in BW and IAT in HFD-fed mice was significantly more potent than that of SP, which is one of the main sericultural products and has a quantitatively and qualitatively similar composition to SMSP (Figure 4B).

It is well known that HFD promotes weight gain in the IAT and liver by converting excess energy into the form of TG, and that the resultant larger IAT and liver lead to higher serum TG levels and hyperglycemia, which is a major risk factor for diabetes and other metabolic disorders, in HFD-fed animals [17,18,19,20,21,22,23,24,25,26,27,28,30,31]. Furthermore, HFD-induced obese mice have increased rates of lipogenesis and cholesterol synthesis, resulting in higher serum cholesterol levels. In the present study, SMSP significantly lowered the levels of serum glucose, TG, TC, and LDLC not only in HFD-fed mice (Table 7) but also in ND-fed mice (Table 3), suggesting that SMSP might be able to ameliorate hyperglycemia and hyperlipidemia in obese patients and prevent obesity and abnormalities in glucose and lipid metabolism in non-obese individuals.

In the HFD-feeding experiment, weekly FI in the SMSP group was often significantly lower than that in the control and orlistat groups, and TFI in the SMSP group was also significantly lower than in both the control and orlistat groups (Figure 3 and Table 5), indicating that the anti-obesity effects of SMSP might be attributed to appetite suppression, while orlistat did not appear to affect appetite in mice. However, FER in each of the orlistat and SMSP groups was significantly lower than in the control group (Table 5), suggesting that the anti-obesity effects of SMSP were not only due to appetite suppression but also to a decrease in the efficiency of converting feed to BW in HFD-fed mice. SP, which has a composition quantitatively and qualitatively similar to SMSP, has been reported to inhibit adipogenesis by blocking adipogenic gene expression [18], leading us to speculate that SMSP might have an effect on adipogenesis similar to that of SP. On the other hand, in the ND-feeding experiment, administration of SMSP resulted in a slight, non-significant dose-dependent decrease in TFI and weekly FI compared to the control group, while FER in the SMSP group was significantly lower than in the control group (Figure 2 and Table 1), suggesting that the anti-obesity effects of SMSP were mainly due to appetite suppression in ND-fed mice.

Elevated levels of liver enzymes are recognized as the first indication of liver disease [13,32]. At the end of each 12-week experiment, serum levels of ALT, AST and ALP in HFD-fed mice were significantly higher than those in ND-fed mice (Appendix A). Administration of SMSP significantly lowered these three serum levels in HFD-fed mice, whereas orlistat had no significant effect on any of these levels compared to controls (Table 8), demonstrating a possible hepatoprotective effect of SMSP against liver damage, which is consistent with previous reports [11,12,13,14]. On the other hand, no significant differences in serum AST, ALT, and ALP levels in ND-fed mice (Table 4) are likely due to the fact that the ND-fed mice in this study were generally healthier than the HFD-fed mice, which is parallel to the situation between non-obese individuals and obese patients.

In summary, our study demonstrated that SMSP efficiently suppressed BWG in both ND-fed and HFD-fed adult mice, which was significant compared to the respective controls from week 2 onwards. At the end of the 12-week ND-feeding experiment, SMSP significantly inhibited FER and weight gain in the IAT with no significant decrease in TFI, implying that the anti-obesity effects of SMSP might be primarily attributed to appetite suppression in ND-fed mice. At the end of the 12-week HFD-feeding experiment, SMSP significantly suppressed weight gain in the IAT and liver and reduced both TFI and FER compared to the respective controls, indicating that SMSP suppressed appetite and simultaneously reduced the conversion of feed into BW in HFD-fed mice. SMSP also significantly lowered serum levels of metabolites related with glucose and lipid metabolism (i.e., glucose, TG, TC, and LDLC) and liver injury markers (i.e., AST, ALT, and ALP) compared to the respective controls. Administration of SMSP had no significant effect on the weights of kidney, spleen, and thymus and on the nephrotoxicity indices (i.e., BUN and creatinine) compared to the respective controls. Taken together, the above results indicate that SMSP has potent anti-obesity effects and is safe for long-term use. SMSP might be a potential therapeutic agent and/or nutraceutical to ameliorate hyperglycemia and hyperlipidemia in obese patients and to prevent obesity and abnormalities in glucose and lipid metabolism in non-obese individuals. Further studies will be needed to determine the minimum effective dose of SMSP for its anti-obesity effects and calculate the equivalent dose in humans, isolate and identify its active component(s), elucidate the mechanisms of its anti-obesity effects through analyses of histopathology and gene expressions, and evaluate its potential as an effective anti-obesity agent for clinical applications in both obese patients and non-obese individuals. In addition, although the present study provided some indicators demonstrating the safety of SMSP, such as serum parameters and intra-abdominal organ weights at necropsy, more comprehensive safety assessments, including its potential long-term side effects, will be necessary.

## Figures and Tables

**Figure 1 foods-12-03613-f001:**
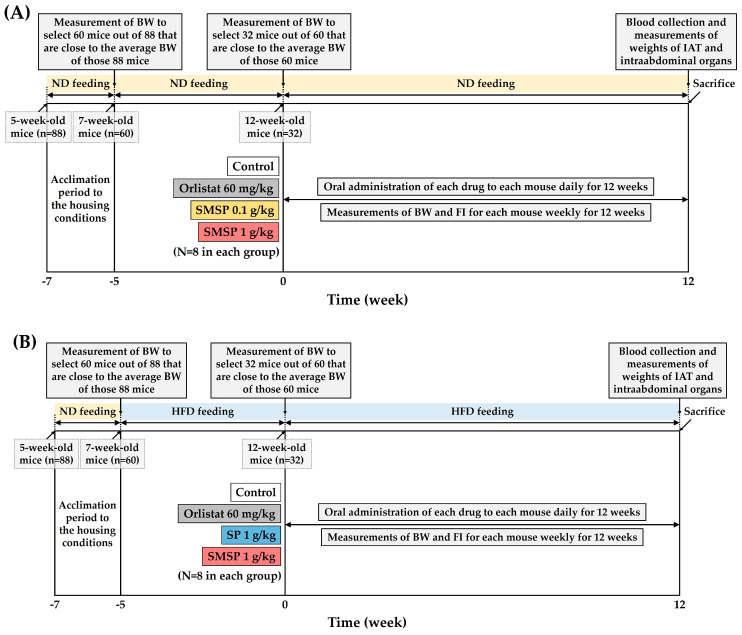
Experimental design to determine the anti-obesity effects of long-term administration of SMSP in ND-fed (**A**) and HFD-fed (**B**) mice, respectively, in vivo.

**Figure 2 foods-12-03613-f002:**
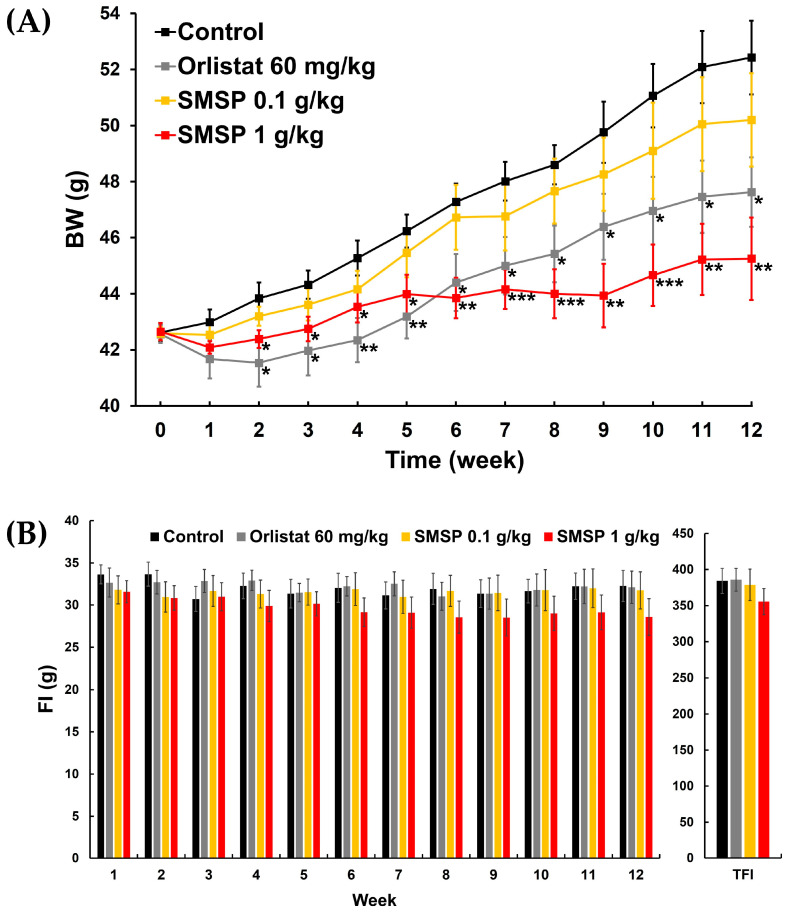
Effects of daily administration of SMSP for 12 weeks on (**A**) changes in BW and (**B**) weekly FI and TFI in ND-fed mice in vivo. Orlistat was used as a positive control drug. Data were evaluated using ANOVA, followed by a Dunnett’s post hoc test, if appropriate. Significant difference (***** *p* < 0.05, ******
*p* < 0.01, and *******
*p* < 0.001) compared to each control value.

**Figure 3 foods-12-03613-f003:**
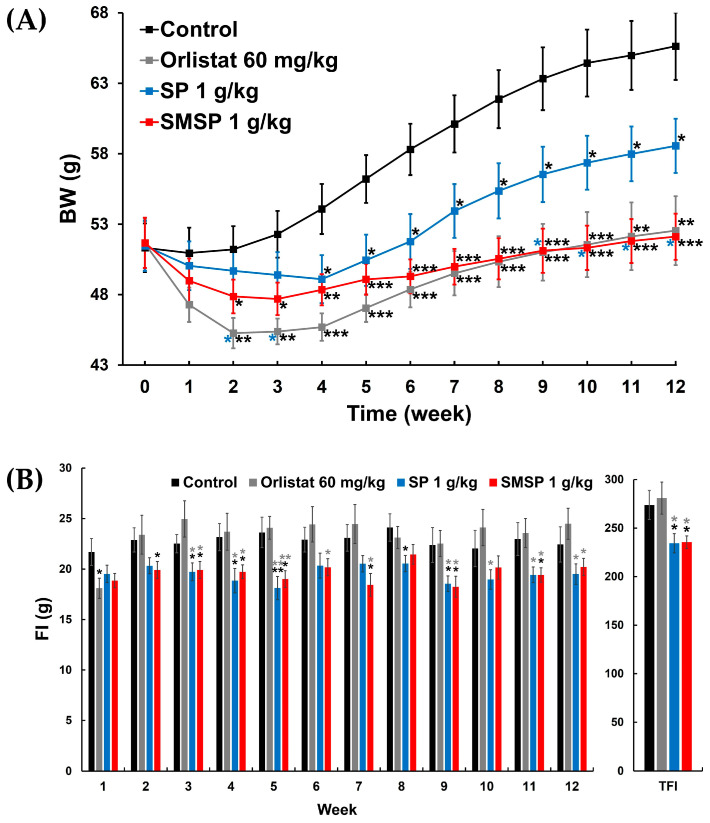
Effects of SP and SMSP administered once daily for 12 weeks, respectively, on (**A**) changes in BW and (**B**) weekly FI and TFI in HFD-fed mice in vivo. Orlistat was used as a positive control drug. * *p* < 0.05, ** *p* < 0.01, and *** *p* < 0.001 versus respective control values (asterisks colored black); * *p* < 0.05 and ** *p* < 0.01 versus respective values in orlistat 60 mg/kg group (asterisks colored gray); * *p* < 0.05 versus respective values in SP 1 g/kg group (asterisks colored blue).

**Figure 4 foods-12-03613-f004:**
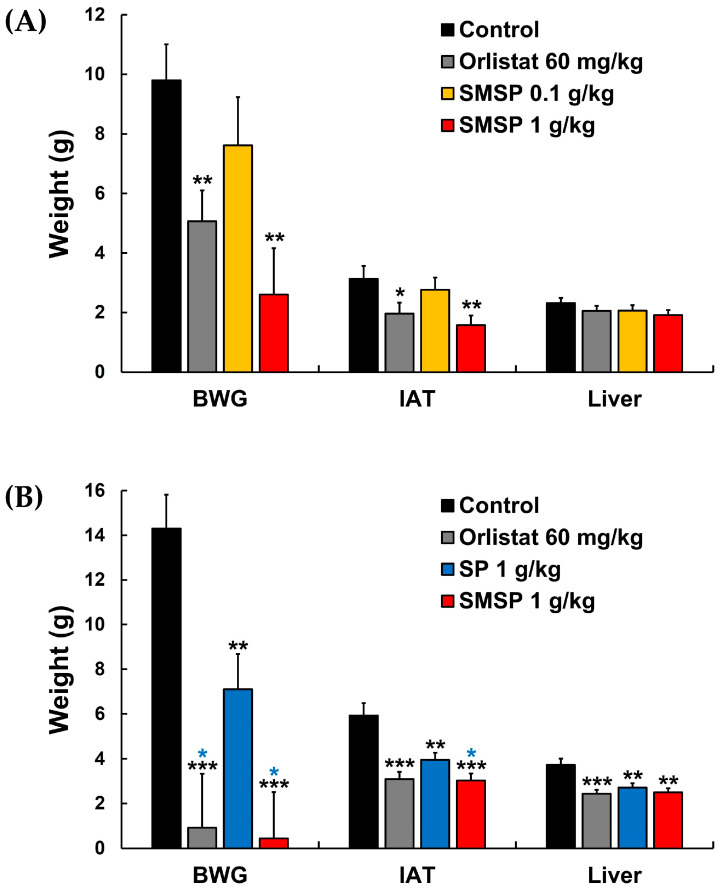
Effects of daily administration of SMSP for 12 weeks in vivo on BWG and weights of IAT and liver in (**A**) ND-fed and (**B**) HFD-fed mice, respectively. ***** *p* < 0.05, ******
*p* < 0.01, and *******
*p* < 0.001 versus respective control values (asterisks colored black); * *p* < 0.05 versus respective values in SP 1 g/kg group (asterisks colored blue).

**Table 1 foods-12-03613-t001:** BWG, TFI, and FER in mice fed ND for 12 weeks.

	Control	Orlistat 60 mg/kg	SMSP 0.1 g/kg	SMSP 1 g/kg
Initial BW (g)	42.63 ± 0.32	42.56 ± 0.31	42.59 ± 0.23	42.65 ± 0.31
Final BW (g)	52.43 ± 1.32	47.63 ± 1.24 *	50.20 ± 1.66	45.25 ± 1.47 **
BWG (g)	9.80 ± 1.21	5.06 ± 1.04 **	7.61 ± 1.62	2.60 ± 1.56 **
TFI (g)	384.23 ± 17.37	385.85 ± 15.83	378.76 ± 21.80	355.49 ± 18.10
FER (%)	2.51 ± 0.25	1.27 ± 0.21 **	1.90 ± 0.35	0.64 ± 0.39 ***

Data are expressed as the mean ± SEM (*n =* 8). * *p* < 0.05, ** *p* < 0.01, and *** *p* < 0.001 versus respective control values.

**Table 2 foods-12-03613-t002:** Weights of IAT and intra-abdominal organs in mice fed ND for 12 weeks.

	Control	Orlistat 60 mg/kg	SMSP 0.1 g/kg	SMSP 1 g/kg
IAT (g)	3.13 ± 0.44	1.97 ± 0.36 *	2.77 ± 0.41	1.57 ± 0.33 **
Liver (g)	2.32 ± 0.18	2.05 ± 0.17	2.06 ± 0.19	1.91 ± 0.17
Kidney (mg)	685.25 ± 50.54	548.63 ± 55.97	611.13 ± 64.26	659.25 ± 53.51
Spleen (mg)	112.13 ± 7.41	98.25 ± 6.36	105.38 ± 7.82	94.63 ± 7.12
Thymus (mg)	31.38 ± 3.64	29.00 ± 3.32	31.50 ± 3.36	30.25 ± 3.81

Data are expressed as the mean ± SEM (*n =* 8). * *p* < 0.05 and ** *p* < 0.01 versus IAT in control group.

**Table 3 foods-12-03613-t003:** Serum glucose and lipid profiles in mice fed ND for 12 weeks.

	Control	Orlistat 60 mg/kg	SMSP 0.1 g/kg	SMSP 1 g/kg
Glucose (mg/dL)	230.14 ± 17.62	167.43 ± 11.64 **	217.14 ± 13.67	160.29 ± 12.40 **
TG (mg/dL)	110.71 ± 7.26	81.33 ± 9.01 *	102.86 ± 5.47	78.17 ± 9.63 *
TC (mg/dL)	192.43 ± 14.89	153.29 ± 10.88 *	185.71 ± 15.15	154.86 ± 11.02 *
HDLC (mg/dL)	147.75 ± 14.66	158.13 ± 14.79	154.13 ± 14.93	163.63 ± 16.27
LDLC (mg/dL)	25.75 ± 3.06	17.13 ± 3.00 *	22.38 ± 2.67	15.75 ± 3.40 *

Data are expressed as the mean ± SEM (*n =* 8). * *p* < 0.05 and ** *p* < 0.01 versus respective control values.

**Table 4 foods-12-03613-t004:** Serum biochemical parameters reflecting renal and hepatic toxicity in mice fed ND for 12 weeks.

	Control	Orlistat 60 mg/kg	SMSP 0.1 g/kg	SMSP 1 g/kg
BUN (mg/dL)	23.50 ± 2.06	22.50 ± 1.65	23.00 ± 1.94	22.25 ± 1.64
Creatinine (mg/dL)	0.22 ± 0.06	0.21 ± 0.04	0.22 ± 0.05	0.20 ± 0.03
AST (IU/L)	93.50 ± 10.02	87.25 ± 7.21	87.88 ± 7.22	82.75 ± 6.02
ALT (IU/L)	27.13 ± 3.51	24.75 ± 3.10	24.00 ± 2.94	22.13 ± 2.87
ALP (IU/L)	48.00 ± 4.02	45.13 ± 3.26	42.88 ± 3.35	41.75 ± 3.26

Data are expressed as the mean ± SEM (*n =* 8).

**Table 5 foods-12-03613-t005:** BWG, TFI, and FER in mice fed HFD for 12 weeks.

	Control	Orlistat 60 mg/kg	SP 1 g/kg	SMSP 1 g/kg
Initial BW (g)	51.33 ± 1.73	51.63 ± 1.76	51.46 ± 1.78	51.68 ± 1.79
Final BW (g)	65.63 ± 2.40	52.54 ± 2.44 **	58.56 ± 1.92 *	52.11 ± 1.64 ***,*
BWG (g)	14.30 ± 1.52	0.91 ± 2.42 ***,*	7.10 ± 1.58 **	0.44 ± 2.07 ***,*
TFI (g)	273.84 ± 15.03	280.94 ± 16.52	234.39 ± 9.87 *,*	235.46 ± 6.46 *,*
FER (%)	5.23 ± 0.50	0.41 ± 0.81 ***,*	3.15 ± 0.69 *	0.27 ± 0.87 ***,*

Data are expressed as the mean ± SEM (*n =* 8). * *p* < 0.05, ** *p* < 0.01, and *** *p* < 0.001 versus respective control values (asterisks colored black); * *p* < 0.05 versus TFI in orlistat 60 mg/kg group (asterisks colored gray); * *p* < 0.05 versus respective values in SP 1 g/kg group (asterisks colored blue).

**Table 6 foods-12-03613-t006:** Weights of IAT and intra-abdominal organs in mice fed HFD for 12 weeks.

	Control	Orlistat 60 mg/kg	SP 1 g/kg	SMSP 1 g/kg
IAT (g)	5.93 ± 0.56	3.09 ± 0.32 ***	3.95 ± 0.32 **	3.02 ± 0.32 ***,*
Liver (g)	3.73 ± 0.28	2.43 ± 0.18 ***	2.71 ± 0.19 **	2.50 ± 0.19 **
Kidney (mg)	726.25 ± 51.70	660.88 ± 39.99	656.75 ± 33.85	635.50 ± 33.96
Spleen (mg)	118.25 ± 11.50	109.50 ± 9.92	112.25 ± 12.46	108.13 ± 10.23
Thymus (mg)	44.50 ± 5.79	35.50 ± 5.58	34.25 ± 6.24	34.50 ± 6.25

Data are expressed as the mean ± SEM (*n =* 8). ** *p* < 0.01 and *** *p* < 0.001 versus respective control values (asterisks colored black); *
*p* < 0.05 versus IAT in SP 1 g/kg group (asterisk colored blue).

**Table 7 foods-12-03613-t007:** Serum glucose and lipid profiles in mice fed HFD for 12 weeks.

	Control	Orlistat 60 mg/kg	SP 1 g/kg	SMSP 1 g/kg
Glucose (mg/dL)	337.50 ± 28.79	238.63 ± 15.98 **	255.00 ± 15.56 *	248.38 ± 15.39 *
TG (mg/dL)	192.00 ± 15.81	130.88 ± 12.43 **	150.63 ± 12.28 *	133.75 ± 11.82 **
TC (mg/dL)	305.57 ± 20.28	219.43 ± 18.63 **	247.14 ± 18.98 *	238.14 ± 17.15 *
HDLC (mg/dL)	163.13 ± 13.89	191.00 ± 17.99	182.88 ± 15.05	199.13 ± 17.30
LDLC (mg/dL)	48.38 ± 5.76	31.63 ± 5.20 *	38.63 ± 5.08	32.13 ± 5.28 *

Data are expressed as the mean ± SEM (*n =* 8). * *p* < 0.05 and ** *p* < 0.01 versus respective control values.

**Table 8 foods-12-03613-t008:** Serum biochemical parameters reflecting renal and hepatic toxicity in mice fed HFD for 12 weeks.

	Control	Orlistat 60 mg/kg	SP 1 g/kg	SMSP 1 g/kg
BUN (mg/dL)	26.13 ± 2.69	25.00 ± 2.19	23.88 ± 2.29	24.50 ± 2.10
Creatinine (mg/dL)	0.25 ± 0.05	0.25 ± 0.04	0.24 ± 0.04	0.24 ± 0.04
AST (IU/L)	155.75 ± 15.99	129.38 ± 17.26	107.63 ± 12.41 *	114.25 ± 12.29 *
ALT (IU/L)	45.88 ± 4.13	39.50 ± 4.94	34.50 ± 5.14	33.63 ± 4.26 *
ALP (IU/L)	68.13 ± 3.83	59.50 ± 4.37	55.13 ± 3.39*	53.25 ± 3.48 **

Data are expressed as the mean ± SEM (*n =* 8). * *p* < 0.05 and ** *p* < 0.01 versus respective control values.

## Data Availability

The data presented in this study are available from the corresponding author upon request.

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
