# Peer review of "Anti-Obesity Effects of the Larval Powder of Steamed and Lyophilized Mature Silkworms in a Newly Designed Adult Mouse Model"

_foods, 2023, doi:10.3390/foods12193613_

Round 1

Reviewer 1 Report

The study evaluated the potential anti-obesity effects of SMSP (Steamed and Lyophilized Mature Silkworm Powder) in adult mice, some of which were fed a normal diet (ND) and others a high-fat diet (HFD). While the study appears promising, there are some limitations and considerations:

Safety Profile: Although the study mentions no significant effects on certain organ weights and blood markers, comprehensive safety assessments, including potential long-term side effects, are necessary.

It is not clear why authors have selected SMSP 0.1g/kg BW for ND group and 1 g/kg BW for HFD group.

Therapeutic Agent: While the study suggests the potential use of SMSP as a nutraceutical or therapeutic agent, regulatory approvals, and further research in humans would be required for such applications.

Authors have mentioned drug throughout the study for SMSP dose, I would suggest to rephrase the word to something like treatment or agent or other suitable rather than drug.

Has authors performed any statistical analysis between control groups from ND and HFD fed animals. It would help to understand the model better.

Reviewer 2 Report

The article titled "Anti-Obesity Effects of Larval Powder of Steamed and Lyophilized Mature Silkworms in A Newly Designed Adult Mouse Model" published in the journal "Foods." Overall, I find the research to be valuable and insightful. The authors have conducted a commendable study; however, there are some minor suggestions for improvement.

-          A table or list of abbreviations and acronyms used in the article would greatly enhance the readability of the paper. It will help readers easily understand the terminology and abbreviations used throughout the text.

-          It would be beneficial to clarify whether the active compounds and insect-based mixtures were administered to the mice along with their standard diet or if the insect preparations served as the primary diet in the experimental groups. Specifying this detail, including dosage in g/kg/day, would provide a clearer understanding of the study design.

-          Given that silkworm larvae are rich in fat, it is essential to explore the chemical composition of the fat fraction in more detail. Investigating the impact of the fat fraction on anti-obesity effects in future studies could provide valuable insights into the mechanisms at play.

-          The article mentions cellulose (lines 217-220) as the fiber component in insects, but it should be noted that insects primarily contain chitin as a form of fiber. This correction will ensure the accuracy of the information presented.

-          The study examines doses of larval powder ranging from 60 mg to 1 g per kg of body weight. To make the results more relevant to potential human applications, it would be beneficial to calculate the average daily dose for an adult human. This calculation can provide insights into the feasibility of using larval powder as a nutraceutical, considering that effective nutraceuticals are typically administered in relatively small quantities.

In conclusion, the article "Anti-Obesity Effects of Larval Powder of Steamed and Lyophilized Mature Silkworms in A Newly Designed Adult Mouse Model" contributes significantly to the understanding of insect-based nutraceuticals and their potential anti-obesity effects. The suggested improvements, such as providing an abbreviation table, clarifying dietary aspects, exploring the fat fraction, correcting the mention of chitin, and calculating human-equivalent dosages, would enhance the article's quality and impact. I recommend that the authors consider these suggestions to further strengthen their research.
